# Efficiency Boost in Dye-Sensitized Solar Cells by Post- Annealing UV-Ozone Treatment of TiO_2_ Mesoporous Layer

**DOI:** 10.3390/ma14164698

**Published:** 2021-08-20

**Authors:** Dariusz Augustowski, Maciej Gala, Paweł Kwaśnicki, Jakub Rysz

**Affiliations:** 1Department of Advanced Materials Engineering, Faculty of Physics, Astronomy and Applied Computer Science, Jagiellonian University, Łojasiewicza 11, 30-348 Cracow, Poland; augustowski@doctoral.uj.edu.pl (D.A.); maciej.gala@doctoral.uj.edu.pl (M.G.); jakub.rysz@uj.edu.pl (J.R.); 2Research & Development Centre for Photovoltaics, ML System S.A. Zaczernie 190G, 36-062 Zaczernie, Poland; 3Department of Physical Chemistry and Physicochemical Basis of Environmental Engineering, Institute of Environmental Engineering in Stalowa Wola, John Paul II Catholic University of Lublin, Kwiatkowskiego 3A, 37-450 Stalowa Wola, Poland

**Keywords:** photovoltaics, solar cells, dye-sensitized solar cells, UV-ozone treatment

## Abstract

The organic residues on titanium(IV) oxide may be a significant factor that decreases the efficiency of dye-sensitized solar cells (DSSC). Here, we suggest the UV-ozone cleaning process to remove impurities from the surface of TiO_2_ nanoparticles before dye-sensitizing. Data obtained from scanning electron microscopy, Kelvin probe, Fourier-transform infrared spectroscopy, and Raman spectroscopy showed that the amounts of organic contamination were successfully reduced. Additionally, the UV-VIS spectrophotometry, spectrofluorometry, and secondary ion mass spectrometry proved that after ozonization, the dyeing process was relevantly enhanced. Due to the removal of organics, the power conversion efficiency (PCE) of the prepared DSSC devices was boosted from 4.59% to 5.89%, which was mostly caused by the increment of short circuit current (*J_sc_*) and slight improvement of the open circuit voltage (*V_oc_*).

## 1. Introduction

The groundbreaking work of M. Grätzel and B. O’Regan [1] in the early 1980s encouraged many scientific groups to undertake research on dye-sensitized solar cells (DSSC) [2]. These photovoltaic devices represent the third generation of solar cells [3], which are not based on the type-specific *p*-*n* junction. DSSC devices have low production costs [4], are environmentally friendly [5], and have the ability to be mass-produced [6]. Such advantages are driving the interests of many research groups. DSSCs are also susceptible to modifications. Thus, there are many scientific papers focused on changing the original materials used by Grätzel (Figure 1a) or modifying them. The mesoporous titanium(IV) oxide layer used originally as a photoanode was exchanged by zinc oxide [7,8] and nickel(II) oxide [9] nanoparticles or doped with tungsten [10], sulfur [11] or copper [12]. Platinum nanoparticles forming the counter electrode were successfully replaced [13] by e.g., carbon [14], tungsten disulfide [15], or molybdenum oxide [16]. A great effort was also made to change the iodine-based electrolyte. As a result, water- [17] or polymer- [18,19] based DSSC electrolytes were demonstrated.

Another reported possibility to improve the dye-sensitized solar cells’ performance was to alter the original internal architecture. The additional scattering layer made out of TiO_2_ particles larger than 200 nm is a good example [20]. This structure scatters the electromagnetic waves, thus increasing the optical path of the light in the active layer. Another way to enhance the power conversion efficiency (PCE) is to attenuate the electron–electrolyte recombination process which occurs mainly at FTO (fluorine-doped tin oxide, SnO_2_:F) and electrolyte interface [21]. To prevent this, electron-blocking layers were developed. Thin continuous layers of metal oxides deposited on the mesoporous TiO_2_ layer [22] or directly on the FTO surface [23] improve DSSC performance. It was proved that many materials, both semiconductors such as ZnO [24], TiO_2_ [25], and MgO [26] as well as insulators such as HfO_2_ [27] or Al_2_O_3_ [27]_,_ can effectively attenuate the electron–electrolyte recombination. Recently, the plasmonic effect in DSSCs was reported [28]. A quantum-sized metal nanoparticle could boost the efficiency of a photovoltaic device by improving the charge transport [29] or the scattering effect [30] inside the solar cell active layer. There are also reported approaches to extend the absorption spectrum in solar cells. When the TiO_2_ layer was sensitized by two or more dyes absorbing in different spectral ranges, the light was harvested more efficiently [31,32]. Hence, the DSSCs were able to generate greater photocurrents and increase their efficiency.

In this article, we point out another aspect that affects the performance of DSSC devices and propose an effortless and cheap procedure to improve it. In most cases, the fabrication of the photoanode is based on screen printing of organic paste with titanium(IV) oxide nanoparticles [33,34]. For that reason, after the sintering process at high temperatures (<600 °C), there are still some organic contaminations [35] within the porous film of TiO_2_. It is difficult to remove them by solvents from the bulk as it could be absorbed deep into the mesoporous structure by capillary action. Then, a long baking process is normally required. The easiest way to reduce the amount of impurities is to apply reactive gas which can easily penetrate any microstructure. Here, we report the results of treating the mesoporous layer of sintered TiO_2_ nanoparticles with UV-ozone cleaner (Figure 1b). 

There are at least a few studies showing that high-power UV-ozone is able to change the surface properties of mesoporous TiO_2_. Reported by Dawo et al. [36] and Saekow et al. [37], results proved that O_3_ molecules led to enhanced adsorption of dye molecules, and thus higher photocurrents in DSSC devices. The authors analyzed a few properties of the modified layers—e.g., stoichiometry, wettability, and roughness. As suggested there, the reduction of Ti^4+^ to Ti^3+^ produced more electrons which could improve the charge transport and reduce the electron–hole recombination process. Enhanced wettability (lower contact angle) may help the dye solution to penetrate deeper within the layer. The authors have also indicated that UV-ozone treatment decreased the roughness (proved by atomic force microscope measurements), which could potentially lead to more efficient harvesting of incident light and better dye adsorption. Furthermore, they have implied that more experiments should be performed to explore the effect of UV-ozone treatment of TiO_2_ layers used in DSSC devices.

Data obtained by us showed the additional effect—removing of organic residuals—led to enhanced adsorption of dye molecules and improved electrical properties of DSSCs. The cleaning effect was directly examined by spectroscopic methods, such as Raman and Fourier-transform infrared (FTIR) spectroscopies, where we received characteristic signals for organic molecules before the UV-ozone treatment. We have also used indirect methods—e.g., scanning electron microscopy (SEM) and Kelvin probe techniques—to prove that the organic residuals were no more present in the TiO_2_ layer. Consequently, more dye molecules were able to bond to the titanium(IV) dioxide particles, which was proved by absorbance and photoluminescence measurements. The UV-ozone treatment resulted in higher photocurrent, and thus greater PCE of prepared dye-sensitized solar cells. 

## 2. Materials and Methods 

### 2.1. Materials

Chemicals, substrates, and components were bought from commercial sources. The FTO (SnO_2_:F) glass (NSG TECTM A7, 6–8 Ω/□) substrates were purchased from Pilkington (Sandomierz, Poland). Titania pastes (18NR-T and 18NR-AO), platinum paste (PT1), di-tetrabutylammonium cis-bis(isothiocyanato)bis(2,2′-bipyridyl-4,4′dicarboxylato)ruthenium(II) (N719 dye), and iodine-based electrolyte (EL-HPE) were purchased from GreatCell Solar (Elanora, Australia). The electrolyte was composed of acetonitrile, valeronitrile, 1-butyl-3-methylimidazolium iodine, 4-tert-butylpyridine, guanidium, thiocyanate, and iodine. Lamination foil was purchased from DuPont Surlyn^®^ (Wilmington, DE, USA). The 99.8% ethanol was bought from Honeywell (Charlotte, NC, USA).

### 2.2. Methods 

The thickness of TiO_2_ mesoporous layers was determined by a stylus profilometer (Bruker DektakXT, Billerica, MA, USA). The scanning electron microscopy images were obtained by Regulus 8230 (Hitachi, Tokyo, Japan) with a secondary electron (SE) detector. The accelerating voltage and the beam current were set to 10 keV and 4.7 μA, respectively. Nicolet iS50 (Thermo Scientific, Waltham, MA, USA) spectrometer equipped with an additional accessory for specular reflection measurements—VeeMAX III (Pike Technologies, Madison, WI, USA)—was used to obtain infrared absorption spectra. Raman spectra were collected by the LabRAM HR Evolution system (Horiba Scientific, Kyoto, Japan) with the He-Ne laser (λ = 633 nm) and an objective of ×100 magnification. The work function (WF) of the TiO_2_ was measured by Kelvin probe (Instytut Fotonowy, Cracow, Poland) with a gold reference electrode. The spectrophotometric measurements were conducted on a V-670 UV-Vis-NIR apparatus (Jasco, Pfungstadt, Germany) with deuterium and halogen lamps. Absorbance spectra were acquired with incident angle of 0° and integrating sphere. The photoluminescence spectra were recorded by spectrofluorometer FS5 (Edinburgh Instruments, Livingston, UK). The depth profiles were obtained by secondary ion mass spectrometry with a time-of-flight mass analyzer (Iontof, Münster, Germany) in dual beam mode. A cesium ion beam (1 keV, Cs^+^, 75 nA) was used to sputter the TiO_2_ surface simultaneously, while a pulsed bismuth cluster beam (Bi_3_^+^, 30 keV, 0.4 pA) was used to analyze the composition of the central part of the milled crater. To avoid the so-called edge-effect edges, the sputter area was set to 500 × 500 µm^2^ and the analyzed area to 250 × 250 µm^2^. Prepared DSSC devices were electrically characterized by CLASS-01 (PV Test Solutions, Wrocław, Poland) under AM1.5 illumination with a light intensity of 100 mW/cm^2^. Solar cells were also analyzed by intensity-modulated photovoltage (IMVS) and intensity-modulated photocurrent (IMPS) spectroscopies. Devices were illuminated with a light of sinusoidal modulated intensity (5–50 Hz), and a source a 530 nm LED (ThorLabs, Newton, NJ, USA) was used. 

### 2.3. Dye-Sensitized Solar Cells Preparation

The FTO was used as a substrate for the photoanode. Glasses were cleaned and degreased by ultrasonification in acetone, DI water, and isopropanol for 5, 10, and 15 min, respectively. The rectangular shape (3 × 5 mm^2^) titanium paste layers were deposited by a screen-printing method. The first layer was printed by 18NR-T paste with an average nanoparticle size of 25 nm and the second layer by 18NR-AO with nanoparticles <450 nm as a scattering layer. Then, the samples were annealed at 565 °C (heating ramp 200 °C/h, hold for 15 min at maximal temperature) to remove the organic compounds. The thickness of TiO_2_ mesoporous layers, obtained by a stylus profilometer, was approximately 7 µm. Subsequently, one part of the samples was transferred to the UV-ozone cleaner, ZoneSem II (Sanyu Co., Ltd., Tokyo, Japan), and cleaned for 20 min. Due to the presence of UV radiation and oxygen molecules, the UV-induced ozone reacts with organic impurities and adsorbed contaminants, then changes them into a carbon oxide gas which is continuously extracted from the reaction chamber by a vacuum pump. In the next step, both reference and UV-ozone-treated samples were immersed in 10^−4^ M ethanolic solution of ruthenium dye (N719) for 24 h at room temperature. After that, to remove any un-adhered molecules, the substrates were rinsed with ethanol and dried with nitrogen steam. To prepare the counter electrodes, the cleaned FTO glass was screen-printed with platinum paste and annealed at high temperature to receive a thin film of Pt nanoparticles. Both electrodes were then sealed by a lamination foil (60 µm) as a spacer. Finally, the DSSC devices were filled with an iodine-based redox electrolyte.

## 3. Results and Discussion

### 3.1. Contamination Effect in Uncleaned TiO_2_ Layer

Selected samples were transferred to the vacuum chamber of the scanning electron microscope system immediately after annealing at 565 °C and the cleaning processes. Obtained images present a typical structure of sintered TiO_2_ nanoparticles [38]. There was no difference in the nanoscale structure of the reference (Figure 2a) and the UV-ozone treated (Figure 2b) samples. To show the presence of contaminants, the surfaces were exposed to the electron beam for two minutes (areas marked by white symbols). Adsorbed on the TiO_2_ surface, organic residuals and atmospheric adsorbates began to aggregate and polymerize when interfered with the electron beam [39]. Irradiated zones became darker and blurred in SEM images. In contrast to Figure 2a, the irradiated area is hardly visible in Figure 2b, which indicates a significant reduction in organic contaminants for the ozonized sample. 

### 3.2. Determination of Organic Residuals Presence on TiO_2_

The Raman and infrared (IR) spectroscopies give subsurface information about the chemical composition of the measured sample, but they differ in the probing depth. The probing depth for the Raman spectroscopy is <1 mm, while the IR is able to characterize material up to <1 cm underneath the surface [40]. Here, we apply both spectroscopies to study the residuals of titania paste after the sintering process and after the additional UV-ozone treatment.

Figure 3 presents FTIR spectra obtained for pure titania paste, annealed paste (sintered mesoporous layer of TiO_2_ nanoparticles), and a mesoporous layer after the UV-ozone treatment. The titania paste was composed of terpineol and titanium(IV) oxide nanoparticles with the addition of other organic components such as solvents, etc. Thus, the received spectrum contained strong peaks for C-H (~2900 cm^−1^), O-H (~1600 cm^−1^) bands, and multipeak region from 800 to 1500 cm^−1^ for C-O, C=C, C-H bending, and stretching bands. After the annealing process, the Ti-O stretching(~900 cm^−1^), O-H stretching (~1600 cm^−1^), and bending (~3500 cm^−1^) bands [41] become visible. Detailed analysis of the range from 2800 to 3000 cm^−1^ wavenumber (right panel of Figure 3) reveals a weak multipeak signal coming from the remaining organic compounds, which is associated with C-H stretching bands [42]. These peaks disappear as an effect of the UV-ozone cleaning.

Raman spectra recorded for titania paste and mesoporous TiO_2_ layer before and after UV-ozone cleaning are presented in Figure 4. The spectrum corresponding to titania paste is a superposition of signal characteristics for anatase TiO_2_ bands located at 144 cm^−1^ (E_g_), 197 cm^−1^ (E_g_), 399 cm^−1^ (B_1g_), 519 cm^−1^ (A_1g_), and 639 cm^−1^ (E_g_) [43]. Additional peaks typical for organic compounds are located at 237 cm^−1^, 448 cm^−1^, and 609 cm^−1^, which may correspond to δ(CC) and υ(CC) vibrations in aliphatic chains. Detailed analysis of the spectra demonstrated additional weak signals corresponding to vibrations at 969 cm^−1^ and 1033 cm^−1^. We associate them with υ(CC) vibrations in alicyclic/aliphatic chains or aromatic rings. These two signals merged into one in the annealed sample (1002 cm^−1^), which may be caused by the conformation change of organic compounds during high temperature (565 °C) treatment. After the UV-ozone cleaning, all signals in this range disappeared. Both IR and Raman spectroscopy showed that the organic impurities were successfully removed from the subsurface of the TiO_2_ mesoporous layer.

### 3.3. The Influence of Ozone Treatment on TiO_2_

Before the dye sensitizing, the bare TiO_2_ layers were characterized by UV-VIS spectroscopy to obtain an optical energy bandgap. Received absorbance data were recalculated to Tauc plot–(αhυ)^1/2^ versus *hυ*, where *α* is an absorption coefficient and *hυ* is a photon energy [44]. The value of the bandgap was determined by extrapolation of the linear region shown in Figure 5a. Calculated energy bandgaps for the sintered and UV-ozone cleaned samples are the same, *E_g_* = 3.22 ± 0.02 eV, and equal to the typical value for anatase titanium(IV) oxide crystals [45]. This proves that the UV-ozone treatment did not affect the TiO_2_ structure and its bulk properties because any changes in crystal structure or bulk stoichiometry would be seen in different *E_g_* values. 

The cleaning effect is also visible in Kelvin probe measurements. The work function (WF) of titanium(IV) oxide varies with many parameters [46]—preparation method, annealing temperature, annealing atmosphere, crystal structure, etc. The typically reported value of work function for stoichiometric and annealed TiO_2_ crystals is equal ~5.0 eV [47]. Measured WF for the sintered TiO_2_ mesoporous layer was equal to 5.56 ± 0.03 eV. After the UV-ozone cleaning, the WF value turned to 5.06 ± 0.02 eV. This effect could not be explained by slight oxidization of the material surface because, according to other research [47], in the case of TiO_2_, an oxidation process results in higher WF. Here, taking into consideration the presented data received from Raman and FTIR spectroscopies, we assume that lower WF is caused by the removal of the organic residuals. Organic molecules presented on the sample surface created an additional layer on the TiO_2_ nanoparticles, which increased the barrier for the ejection of electrons from the sample. The higher barrier, the greater voltage is needed to eject the electric charge from the material. Furthermore, the oxidation effect (slight change of surface stoichiometry) is more relevant when the titanium(IV) oxide is treated by oxygen plasma [48] rather than a low-power UV-ozone.

### 3.4. Improving the Dye-Sensitizing Efficiency

The TiO_2_ layers sensitized by N719 molecules were characterized by spectrophotometric measurements. Spectra presented in Figure 6a show higher absorption for the UV-ozone treated samples. Similarly, the photoluminescence (PL) spectra show a higher signal under excitation at λ_ex_ = 530 nm for the samples UV-ozone cleaned prior to sensibilization (Figure 6b).

To demonstrate that N719 molecules also had higher adsorption effects in a subsurface layer, the secondary ion mass spectrometry with the time-of-flight analyzer (ToF-SIMS) was used. Obtained depth profiles are presented in Figure 7. The NCS^−^ ions were chosen as a fingerprint of the N719 molecule and their intensity was analyzed as a function of sputter time. Received data showed that more dye molecules were able to adsorb deeper within the structure after the UV-ozone treatment, which was related to the higher signal intensity in the time range from 50 to 600 s. After that time, both signals reached the plateau.

### 3.5. Performance of DSSC Devices

Prepared as described above, dye-sensitized solar cells were characterized by current–voltage measurements under a solar simulator. The results were averaged for five samples for both the reference and solar cells based on a UV-ozone treated TiO_2_ layer. As shown in Figure 8, the UV-ozone treated DSSCs had higher short-circuit currents (*J_sc_*). The ozonization led to a slight increase in the open circuit voltage (*V_oc_*) and the fill factor (*FF*). As the consequence, the PCE was enhanced from 4.59 ± 0.45% to 5.89 ± 0.38% (see Table 1). The value of *FF* is strictly related with the series (*R_s_*) and shunt resistance (*R_sh_*) parameters. We make an assumption that decreased *R_s_* (from 58.1 ± 4.3 Ω to 49.1 ± 3.9 Ω) and increased *R_sh_* (from 22.0 ± 5.3 Ω to 39.0 ± 4.8 Ω) were a consequence of reduced recombination processes in solar cells. Hence, it was easier for photogenerated charge carriers to reach the appropriate electrodes before they recombined shortly after excitation when the TiO_2_ and N719 interface was cleansed of organic residuals. In the case of *J_sc_*, the augmented photocurrent should be related with greater adsorption of dye molecules. After the UV-ozone treatment, more molecules were able to bond chemically to the TiO_2_ structure so the incident light (photon flux) could be harvested more effectively to produce more photocarriers. As shown on depth profiles from SIMS measurements, the dye molecules were also adsorbed deeper into the structure. The increased amount of N719 near the TiO_2_ and electrolyte interface may also be due to decreasing in the *R_s_* parameter because greater amounts of generated carriers had a shorter way to the counter electrode. 

The photovoltaics devices were also characterized by intensity-modulated photovoltage (IMVS) and intensity modulated photocurrent (IMPS) spectroscopies. Both methods were used to investigate the charge–recombination time (*τ_rec_*) and transport time (*τ_tr_*) under open circuit and short-circuit conditions, respectively [49]. The transfer function *H* was calculated from relations:(1)$$H_{IMPS}\left(f\right)=\frac{\Delta I}{\Delta \Phi \;\;}e^{i\phi \left(f\right)},$$
(2)$$H_{IMVS}\left(f\right)=\frac{\Delta V}{\Delta \Phi \;\;}e^{i\phi \left(f\right)},$$
where *f* is the frequency, Δ*I*—generated photocurrent, Δ*V*—generated photovoltage, Δ*Φ*—intensity of incident light, and *φ*—phase shift that represents the delay time of the response. The IMVS and IMPS plots were presented in the complex plane (Figure 9) according to equations:(3)$$H_{IMPS}\left(f\right)=\frac{\Delta I}{\Delta \Phi \;\;}(\cos\left(\phi \left(f\right)\right)+i\;sin\left(\phi \left(f\right)\right)=H_{IMPS}{}^{\prime }\left(f\right)+i\;H_{IMPS}{}^{\prime \prime }\left(f\right)$$
(4)$$H_{IMVS}\left(f\right)=\frac{\Delta V}{\Delta \Phi \;\;}(\cos\left(\phi \left(f\right)\right)+i\;sin\left(\phi \left(f\right)\right)=H_{IMVS}{}^{\prime }\left(f\right)+i\;H_{IMVS}{}^{\prime \prime }\left(f\right)$$

The recombination time was obtained from the frequency of the point that corresponds to the minimum value of the $H_{IMVS}{}^{\prime \prime }$ (*f_IMVS_*) and the charge–transport time from the frequency of the point that corresponds to the maximum of the $H_{IMPS}{}^{\prime \prime }$ (*f_IMPS_*):(5)$$\tau_{rec}=\frac{1}{2\pi f_{IMVS}\;\;}$$
(6)$$\tau_{tr}=\frac{1}{2\pi f_{IMPS}\;}$$

Summarized in Table 2, the parameters showed that the localization of extrema for IMVS and IMPS plots are different for reference and solar cells based on ozone-cleaned TiO_2_ layers. The calculated values of recombination time were lower for UV-ozone-treated samples (7.3 ± 0.3 ms) than for the reference (16.6 ± 0.8 ms), while the transport time decreased from 12.0 ± 0.9 ms to 8.6 ± 0.3 ms. These findings are in accord with the change in *R_s_* that decreased for ozonized samples. Based on those parameters, we can assume that the cleaning of organic residues from the TiO_2_ mesoporous layer improves the current flow through the DSSC by decreasing the barriers on the titanium(IV) oxide/electrolyte interface.

## 4. Conclusions

UV-ozone cleaning was studied as an efficient method to remove organic residuals and adsorbates from the TiO_2_ structure in dye-sensitized solar cells. A short, 20 min long process of low-power ozonization led to a significant reduction in the amount of contaminants. The cleaning effect was proved by surface and subsurface sensitive methods. The signals corresponding to C-H and C-C vibrations in FTIR and Raman spectroscopy were completely diminished. The additional effect of UV-ozone treatment was the improvement of the dye-sensitizing process. Based on spectroscopic and spectrometric measurements, we showed that the amount of adsorbed N719 dye molecules was relatively augmented. Solar cell devices were characterized by current–voltage measurements and intensity-modulated photovoltage and photocurrent spectroscopies. Obtained data showed that the main factor increasing the DSSCs PCE was a boost in the *J_sc_*, a slight improvement of *V_oc_*, and *FF*. Finally, the efficiency of the prepared solar cells was enhanced from 4.59% to 5.89% with the application of UV-ozone treatment of the TiO_2_ layer.

## Figures and Tables

**Figure 1 materials-14-04698-f001:**
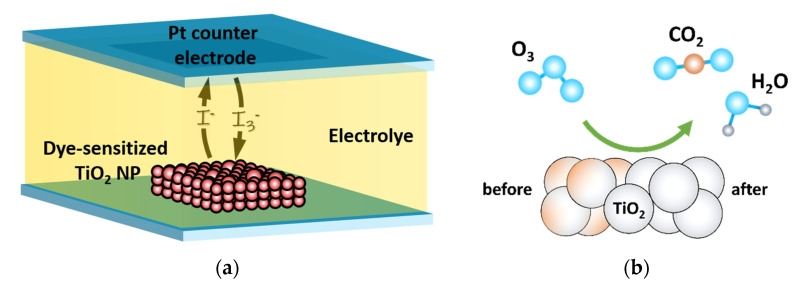
Schematic representation of (**a**) DSSC architecture and (**b**) UV-ozone cleaning process.

**Figure 2 materials-14-04698-f002:**
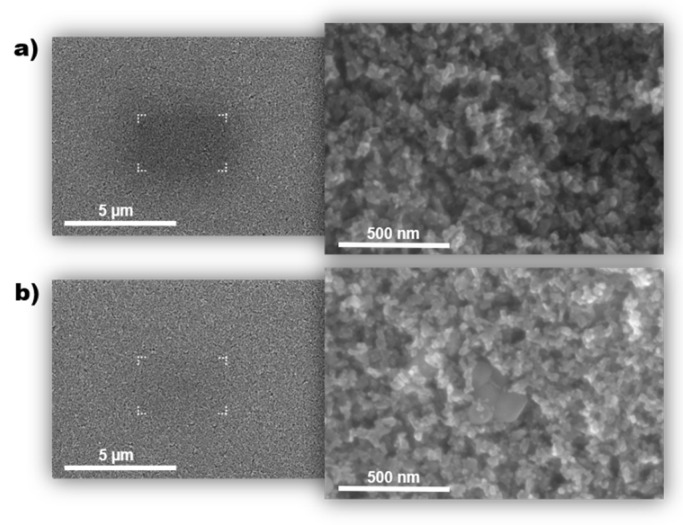
Scanning electron microscopy images of (**a**) reference and (**b**) UV-ozone treated TiO_2_ mesoporous layer. Marked areas indicate the presence of contaminations on the surface.

**Figure 3 materials-14-04698-f003:**
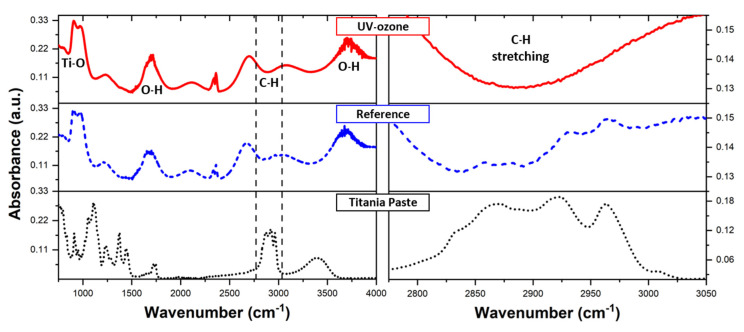
FTIR spectra of titania paste (dotted black) and mesoporous TiO_2_ layer before (dashed blue) and after (solid red) UV-ozone cleaning.

**Figure 4 materials-14-04698-f004:**
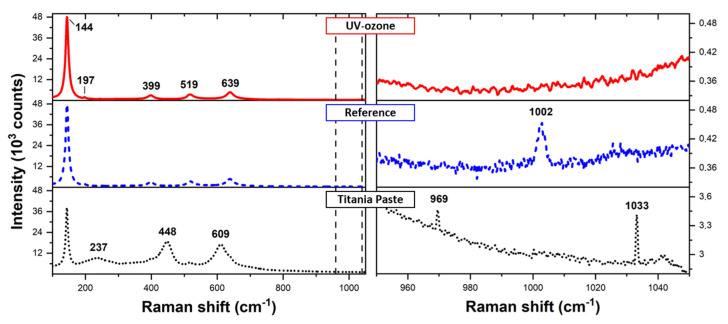
Raman spectra of titania paste (dotted black) and mesoporous TiO_2_ layer before (dashed blue) and after (solid red) UV-ozone cleaning.

**Figure 5 materials-14-04698-f005:**
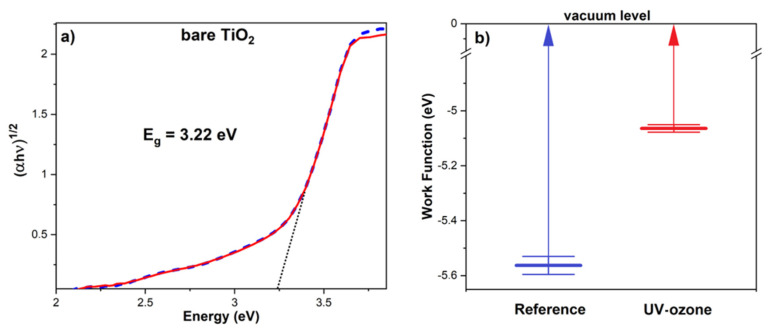
(**a**) Tauc plot of bare TiO_2_ with calculated bandgap before (dashed blue) and after (solid red) UV-ozone cleaning and (**b**) work functions (WF) obtained by Kelvin probe.

**Figure 6 materials-14-04698-f006:**
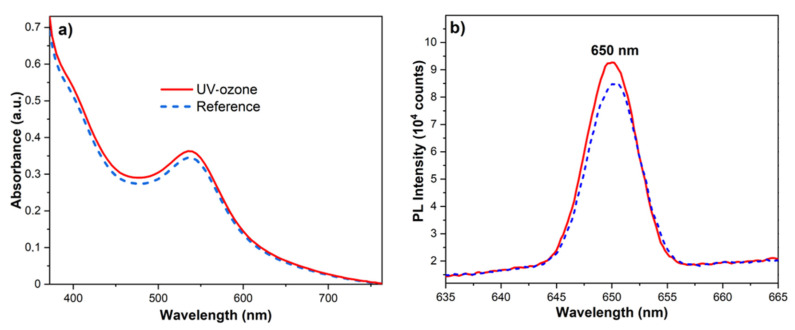
(**a**) Absorbance and (**b**) photoluminescence spectra of dye-sensitized TiO_2_ before (dashed blue) and after (solid red) UV-ozone treatment.

**Figure 7 materials-14-04698-f007:**
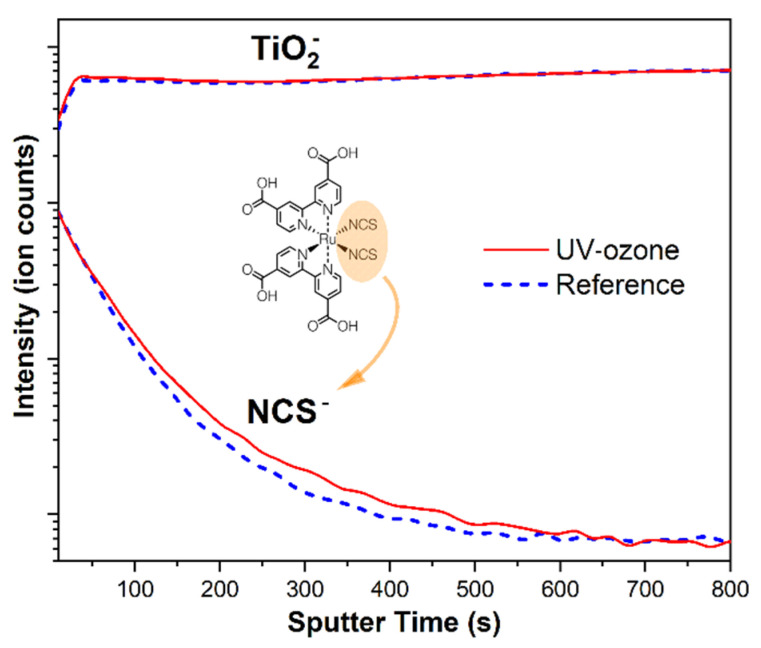
SIMS depth profiles of dye-sensitized TiO_2_ for sintered (dashed blue) and UV-ozone treated (solid red) samples. The analyzed ions were: NCS^−^ for N719 molecule and TiO_2_^−^ for titanium(IV) oxide nanoparticles.

**Figure 8 materials-14-04698-f008:**
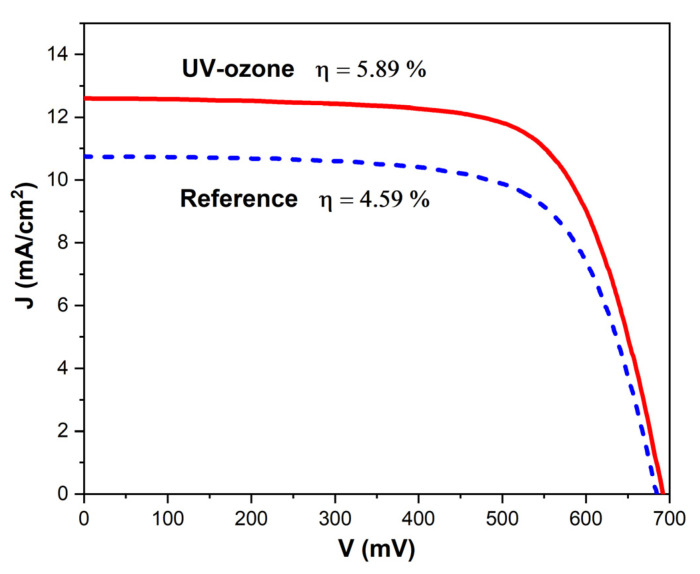
Current–voltage characteristics of reference (dashed blue) and UV-ozone treated (solid red) DSSCs.

**Figure 9 materials-14-04698-f009:**
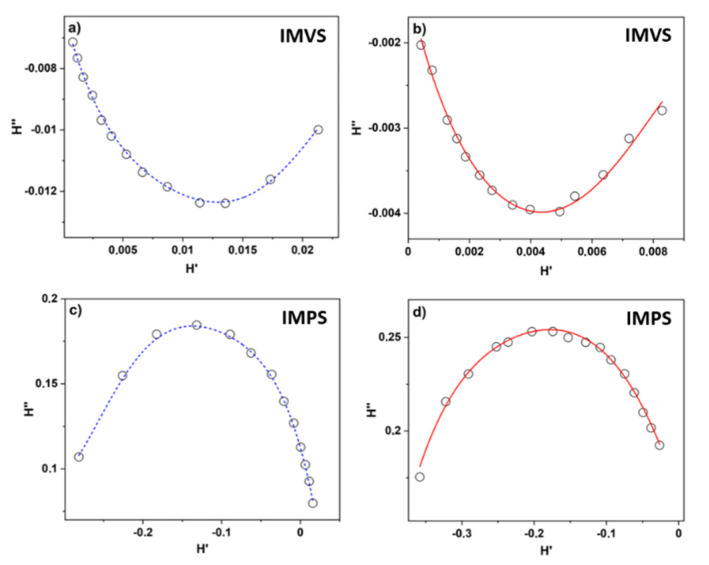
The IMVS (**a**,**b**) and IMPS (**c**,**d**) responses for the reference (**a**,**c**) (dashed blue) and UV-ozone treated (**b**,**d**) (solid red) DSSCs illuminated with LED diode (λ = 530 nm).

**Table 1 materials-14-04698-t001:** The DSSCs parameters obtained by current–voltage characteristics.

Sample	*V_oc_* (mV)	*J_sc_* (mA/cm^2^)	*FF*	PCE (%)	*R_s_* (Ω)	*R_sh_* (kΩ)
Reference	667 ± 10	10.3 ± 1.0	0.67 ± 0.01	4.59 ± 0.45	58.1 ± 4.3	22.0 ± 5.3
UV-ozone	679 ± 10	12.3 ± 0.7	0.70 ± 0.01	5.89 ± 0.38	49.1 ± 3.9	39.0 ± 4.8

**Table 2 materials-14-04698-t002:** Parameters obtained by intensity-modulated photovoltage (IMVS) and intensity-modulated photocurrent (IMPS) spectroscopies for reference and UV-ozone treated DSSCs.

Sample	*f_IMVS_* (Hz)	*τ_rec_* (ms)	*f_IMPS_* (Hz)	*τ_tr_* (ms)
Reference	9.6 ± 0.5	16.6 ± 0.8	13.3 ± 0.2	12.0 ± 0.9
UV-ozone	21.9 ± 0.3	7.3 ± 0.3	18.5 ± 0.7	8.6 ± 0.3

## Data Availability

The data is available within the manuscript.

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
