# Peer review of "Efficiency Boost in Dye-Sensitized Solar Cells by Post- Annealing UV-Ozone Treatment of TiO2 Mesoporous Layer"

_materials, 2021, doi:10.3390/ma14164698_

Round 1
Reviewer 1 Report
The authors report the study of UV-ozone treatment of the TiO2 mesoporous layer in dye sensitized solar cells and a resulting increase in device efficiency. This treatment removes organic contaminants on the TiO2 porous surfaces, as evidenced by FTIR, photoluminescence, SEM, Raman, and UV-Vis spectroscopy measurements of materials. Device performance characteristics are measured using current-voltage as well as intensity modulated photovoltage and intensity modulated photocurrent spectroscopies. Most information is reported in a way that clear to understand by the reader, although some additional information is needed prior to publication.
The Introduction should include a specific discussion of the efficiencies of this type of solar cell device structure and note how exactly it was improved upon or greater understanding obtained by the present study.
In Section 2.2 Methods, define “SE” as used before “detector”
Please review all text for grammatical and technical correctness. There are some minor wording issues that should be corrected. For example, the sentence “Hardly, the contamination can be washed away by any solvent as it may be capillary sucked deep into the structure and there could be difficulty in removing it from the bulk, a costly and long-lasting baking process needs to be used.” Should be separated into smaller clearer sentences such as “It is difficult to remove contamination by solvents from the bulk as it is absorbed deep into the mesoporous structure by capillary action. A long baking process is normally required.”
Also, note the annealing time at 565 C in Section 2.3.
Figures 3ab, 4ab, and 6b provide numerical values for the y-axis and note the offsets fore each data set. Ideally all data would be shown on the same scale, however, if data sets were normalized to a given value then this axis labels and figure captions should note that the data sets were normalized.
More details about the measurement and data reduction process to determine the absorption coefficient used in the Tauc plot (Figure 5a) are required. Note if UV-VIS spectroscopy reflection, transmission, or both were used and the specific formalism used to obtain the absorption coefficient as well as any other measurement configurations (angle of incidence, specular, diffuse, total hemispherical, etc.). Numerical values for the absorption coefficient or the y-axis of the Tauc plot are required so that the reader can identify the range over which extrapolation is performed for comparison with their own results as well as assess the amount of sub-gap absorption evident in Figure 5a.
Please elaborate the discussion of the specific improvements in device performance as evidenced from the JV, IMVS, and IPMS measurements. For example, why did the short circuit current density increase from removal of the organics? There is not a substantial difference in the absorbance between the treated and untreated samples (Figure 6a), so this behavior may have a less obvious origin. Linking performance parameter increases directly and explicitly with material characteristics after UV ozone treatment is crucial to highlighting the impact of this work for the benefit of the readers. I recommend adding a paragraph explaining the reasons why each device performance parameter changed to the extent observed as supported by the other experimental measurements as needed.
Reviewer 2 Report
This paper describes the use of ozone and ultraviolet light to boost the performance of TiO2 based dye-sensitized solar cell via removal of residual organic material. Ample of characterization measurements are provided and results obtained are scientific and sound. However, the idea presented (use of UV-radiation + ozone to enhance performance of dye-sensitized solar cells made from TiO2) is not new. The authors makes no justification for what is novel or new in their approach nor do they cite important previous works on the very same topic (see f. ex. Saekow et al., ”High intensity UV radiation ozone treatment of nanocrystalline TiO2 layers for high efficiency of dye-sensitized solar cells”, Journal of non-crystalline solids, 358 (2012), and more recently; Dawo et al., “Effect of UV-ozone exposure on the dye-sensitized solar cells performance”, Solar energy 208 (2020)).
Additonal minor comments: Many grammatical errors. Excessive use of definite article.
Reviewer 3 Report
The paper entitled “Efficiency Boost in Dye-Sensitized Solar Cells by Post-Annealing UV-ozone treatment of TiO2 Mesoporous Layer” is submitted in Materials. The paper looks interesting and may benefit the research community. However, I feel it can be further improved, after addressing the following suggestions and comments.
- There is a significant decline in the work function from 5.56 eV to 5.06 eV after UV-ozone treatment (line #178-180). The explanation provided for this trend is not convincing for the reviewer. Please discuss it thoroughly.
- Please define acronyms or abbreviations when first used in the abstract/text. After defining, please don’t use full name. For example, power conversion efficiency (PCE) at lines #44, 65, 253, 210, and so on.
- Please check the typos. Line#21, (dyeing), line#69 (SnO2) etc.
- It is suggested to provide enough explanation regarding improvement in FF values after UV-ozone treatment (Table 1). How does this treatment influence the FF values?
- Among various reported techniques to improve the performance of DSSCs, the cosensitization method is not mentioned in the introduction. Please include up-to-date studies, https://doi.org/10.1007/s12034-021-02365-x; https://doi.org/10.1016/j.dyepig.2021.109624;
- It is suggested to use short-circuit current density (JSC) for ISC and open-circuit voltage (VOC) for UOC. Apply these terminologies in the text as well as in related Figures and Tables.
- Be consistent in using terminologies. For example, use “Figure” instead of Fig. for all images. Please check, Fig. 1a, Fig. 2a, Fig. 2b, Fig. 6a, Fig. 6b.
- Please elaborate on the bulk properties, mentioned at line#171.
- Please provide the complete composition of iodide electrolyte in section 2.1.
Round 2
Reviewer 2 Report
In their reply, the authors provide some arguments for novelty but only a part of it were added to the manuscript. A more detailed text should be added also to the manuscript, describing what is different from previous (similar) studies.
The grammar still needs to be improved.
Reviewer 3 Report
The authors have responded to all the comments appropriately. The manuscript is now suitable for publication in 'Materials'.
Author Response
Dear Reviewer,
We appreciate that you have found time to assess our manuscript. We are also convinced that after your suggestions the article became more professional and transparent.
Best regards,
Authors